# Development of a Dual Fluorescent Microsphere Immunological Assay for Detection of Pseudorabies Virus gE and gB IgG Antibodies

**DOI:** 10.3390/v12090912

**Published:** 2020-08-20

**Authors:** Chihai Ji, Yingfang Wei, Jingyu Wang, Yuchen Zeng, Haoming Pan, Guan Liang, Jun Ma, Lang Gong, Wei Zhang, Guihong Zhang, Heng Wang

**Affiliations:** 1Guangdong Provincial Key Laboratory of Prevention and Control for Severe Clinical Animal Diseases, College of Veterinary Medicine, South China Agricultural University, Guangzhou 510642, China; jichihai@163.com (C.J.); yingfangwei111@163.com (Y.W.); wju895929167@163.com (J.W.); mkfdz@outlook.com (Y.Z.); phm_scau@163.com (H.P.); liangguanscau@163.com (G.L.); 18814116634@163.com (J.M.); gonglang@scau.edu.cn (L.G.); 2Key Laboratory of Zoonosis Prevention and Control of Guangdong Province, Guangzhou 510642, China; zw3222009@126.com; 3Bioland Laboratory (Guangzhou Regenerative Medicine and Health Guangdong Laboratory), Guangzhou 510642, China; 4Guangdong Laboratory for Lingnan Modern Agriculture, Guangzhou 510642, China; 5National Engineering Research Center for Breeding Swine Industry, Guangzhou 510642, China

**Keywords:** pseudorabies virus, fluorescent-microsphere immunoassays, gE protein, gB protein, specificity, sensitivity

## Abstract

Pseudorabies, also known as *Aujezsky’s* disease, is an acute viral infection caused by pseudorabies virus (PRV). Swine are one of the natural hosts of pseudorabies and the disease causes huge economic losses in the pig industry. The establishment of a differential diagnosis technique that can distinguish between wild-type infection and vaccinated responses and monitor vaccine-induced immunoglobulin G(IgG) is crucial for the eventual eradication of pseudorabies. The aim of this study was to develop a rapid dual detection method for PRV gE and gB protein IgG antibodies with high specificity and sensitivity. PRV gE codons at amino acid residues (aa) 52–238 and gB codons at aa 539–741 were expressed to obtain recombinant PRV gE and gB proteins via a pMAL-c5x vector. After purification with Qiagen Ni–nitrilotriacetic acid (NTA) agarose affinity chromatography, the two proteins were analyzed via SDS-PAGE and immunoblotting assays. Two single fluorescent-microsphere immunoassays (FMIAs) were established by coupling two recombinant proteins (gE and gB) to magnetic microbeads, and an effective dual FMIA was developed by integrating the two single assays. Optimal serum dilution for each assay, correlation with other common swine virus-positive sera, and comparison with ELISA for two PRV antigens were tested for validation. Compared with ELISA, the specificity and sensitivity were 99.26% and 92.3% for gE IgG antibody detection, and 95.74% and 96.3% for the gB IgG antibody detection via dual FMIA. We provide a new method for monitoring PRV protective antibodies in vaccinated pigs and differentiating wild-type PRV infection from vaccinated responses simultaneously.

## 1. Introduction

Pseudorabies is an important infectious disease in the pig industry that was first described by Aladár Aujeszky in the 1900s and identified in 1930s. It causes reproductive failure in sows, high mortality of newborn piglets, and respiratory symptoms in fattening and growing pigs [1]. A 2011 outbreak of severe pseudorabies in Northern China caused huge economic losses and spread rapidly to large numbers of vaccinated pig farms throughout the country [2,3]. Increased virulence was identified during the epidemic, and commonly used vaccines provided little protection against the pathogen [4,5].

Pseudorabies virus (PRV) is a member of the *Alphaherpesvirinae* subfamily that belongs to the *Herpesvirdae* family [6]. PRV is an enveloped, double-stranded DNA virus with a genome of approximately 150 kb that encodes about 70 proteins [7]. The viral envelope glycoprotein plays an important role in immunity and virus–cell interaction against PRV infection [8,9], and has therefore been one of the hotspots of PRV research. To date, 11 PRV glycoproteins have been identified; among these, gB, gC, and gD are the main protective antigens and they stimulate the production of neutralizing antibodies and virus-specific cellular immune responses in pigs [10]. Mouse and pig serum responses induced by PRV gB protein can neutralize virus infection in vitro, and gB protein can protect both mice and pigs from PRV infection after immunization [11]. The gE protein is an important antigen for virulence and viral (neuronal) spread. It can induce homotypic T-cell aggregation and transmit the virus to highly susceptible cells by activating extracellular signal-regulated kinase 1/2 signaling pathways [12,13]. A major epitope in the gE amino acid (aa) 52–238 domain and eight epitopes in the gB aa 539–741 domain have been identified [14,15]. In general, vaccination is a valuable method for the control of PRV. The advantage of using PRV gE protein as a marker to differentiate wild-type from vaccine-induced PRV responses is that the gE gene is expressed in all wild-type strains, and the gE-deficient vaccine is safer than other gene-deficient vaccines [16,17,18]. Some European countries and North America have eradicated PRV from domestic pigs by using the gE-deleted vaccine [19,20]. The use of a gene deletion vaccine combined with a differential diagnosis method can effectively distinguish vaccinated animals from those with wild-type infection, and is an important measure for the prevention and control of pseudorabies. Therefore, we used gE and gB proteins as diagnostic antigens in the present study.

Microsphere technology with microspheres of two different sizes was used to analyze two antibodies via flow cytometry in the 1970s [21]. A fluorescent-microsphere immunoassay (FMIA) is a new clinical laboratory diagnostic method with high throughput, sensitivity, and specificity for the simultaneous detection of several analytes in complex samples [22]. FMIA can also detect multiple antigens in a single pathogen. Luminex technology, which uses colored magnetic polystyrene beads to capture antigens, is well known and widely applied for the detection of analytes [23]. Because of its multiplexing capability, this method has been used for differentiating infected from vaccinated animals for viral pathogens such as foot and mouth disease virus [24], West Nile virus [25], avian influenza virus [26], and Rift Valley fever virus [27]. ELISA is a traditional diagnostic method for detecting a viral antigen or antibody [28]. Currently, ELISA is used to detect PRV gE and gB IgG antibodies for monitoring the level of viral infection and protective antibodies in clinical diagnosis [29,30]. One disadvantage of ELISA is that it usually detects only a single target molecule and involves high costs when more than one molecule needs to be detected. The development of an efficient diagnostic technique for monitoring protective antibodies in vaccinated pigs and differentiating PRV wild-type infection from vaccinated animals provides an important basis for the prevention and control of PRV infection. Compared with ELISA, FMIA has the following advantages: (1) high throughput, with the detection of 500 indexes of a sample or a reaction system simultaneously; (2) high sensitivity, with the detection of tiny amounts of target substances (minimum detection concentration of 0.01 pg/mL); (3) high accuracy, which needs 100 microspheres to detect each sample to determine the fluorescence value; and (4) cost and time savings compared with traditional ELISA when more than three targets need to be detected simultaneously. However, one limitation is the high initial investment cost, because the Luminex flex Map 3D detection platform is more expensive than an optical density reader. Fluorescent microspheres are protein carriers with good performance to which an antigen or antibody can be coupled for the specific detection of the corresponding antibody or antigen. Therefore, they are often used in immunoassays and diagnostic kits.

We generated gE-12 and gB-28 microsphere complexes by coupling recombinant PRV gE and gB proteins to Number 12 and 28 magnetic microbeads. We used these to develop a double fluorescence-microsphere immunological method for the detection of gE and gB protein IgG antibodies. We compared results from our assay with ELISA results. Our research provides a novel method for the simultaneous detection of PRV gE and gB protein IgG antibodies in clinical practice, and discrimination between PRV-infected and vaccinated animals.

## 2. Materials and Methods

### 2.1. Virus Strain, Serum Samples, and Antibodies

The PRV Guangdong strain was preserved in our laboratory; the complete genome sequence is available in National Center for Biotechnology Information(NCBI)GenBank (KU056477.1/KT948041.1). Negative sera and antiserum against PRV gE/gB, classical swine fever virus (CSFV), porcine circovirus (PCV), and porcine reproductive and respiratory syndrome virus (PRRSV), were preserved in the Diagnostic Laboratory, College of Veterinary Medicine, South China Agricultural University, and stored at −80 °C. IRDye 800CW goat anti-mouse IgG (H + L) was purchased from LI-COR Biosciences (Lincoln, Nebraska, United States of America). Rabbit F (ab) anti-pig IgG H&L (Biotin) and goat anti-mouse IgG H&L (Biotin) were obtained from Abcam (Cambridge, Massachusetts, United States of America). Micro Plex^®^-C Microspheres (Number 12 and 28 magnetic microbeads) were purchased from Luminex (Seattle, Washington, United States of America). Microplates (96-well) were purchased from Greiner (Shanghai, China).

### 2.2. Expression of Recombinant PRV gE and gB Protein

Primers for amplifying the gB and gE encoding regions were designed using Oligo 7.0. Strand-specific primer sequences were synthesized as follows (5′-3′): (gE-F1: GACCATGCGGCCCTTTCTGCT, gE-R1: CCATTCGTCACTTCCGGTTTCTCC, gE-F2: GCGGAATTCATGGAGGCCGACGACGATGAC, gE-R2: GGGAAGCTTTCAATGATGATGATGATGATGGGGCGAGAAGAGCTGCGA, gB-F1: GACAAGCCCGAGGTGTAC, gB-R1: TGGAAGAAGTTGGCGATG, gB-F2: GCGGAATTCATGGACCACATCCAGGCGCAC, gB-R2: GGGAAGCTTTCAATGATGATGATGATGATGGTAGAACTTGAGCGCGTG). After amplification via nested PCR, the gB and gE genes from PRV were cloned in the pMAL-c5x vector to obtain the pMAL-c5x PRV gE and pMAL-c5x PRV gB plasmids. Both recombinant plasmids were identified by restriction endonucleases *EcoR* I and *Hind* III. The plasmids were transformed into *E. coli* DH5α, and the expression was induced with 0.3 mM Isopropyl β-D-Thiogalactoside (IPTG) in an orbital incubator at 16 °C for 8 h.

The induced bacterial solution was centrifuged at 8000× *g* for 10 min, the supernatant was discarded, and bacteria were resuspended in phosphate buffer saline (PBS) (pH 7.0). After ultrasonic disruption (pulses delivered for 3 sec on and 5 sec off over a working cycle of 8 sec), the products were incubated on ice until the solution became clear. The bacterial solution was centrifuged at 12,000× *g* for 20 min at 4 °C. The supernatant was collected and the precipitate containing the recombinant protein was resuspended in PBS for the analysis of protein solubility.

Proteins were purified via Qiagen (Uppsala, Sweden) Ni–NTA agarose affinity chromatography for native His-tagged proteins. Purified proteins were analyzed by SDS-PAGE. After staining with 0.025% Coomassie Blue, the protein bands were visualized. For immunoblotting, the proteins were transferred to polyvinylidene fluoride membranes. After blocking with 5% skimmed milk powder at 37 °C for 2 h, the membranes were incubated with PRV gE/gB positive serum at 4 °C for 8 h and with goat anti-mouse IgG at room temperature for 1 h. The bands visualized with 3,3′,5,5′-tetramethylbenzidine substrate were analyzed by Western blotting (Amersham Biosciences, San Diego, CA, United States of America).

### 2.3. Coupling of Recombinant Proteins to Fluorescent-Encoded Microspheres

Coupling was carried out via a two-step amide reaction (Luminex xMAP). Number 12 and 28 magnetic microbeads were coupled with PRV gE and gB antigens by adding NaH_2_PO_4_ buffer, Sulfo-NHS (N-hydroxysulfosuccinimie sodium salt-sulfo-NHS) and EDC (1-thyl-3-(3-dimethylaminoproy) carbodimide hydrochloride) solutions. The phosphorylation of fluorescent-encoded microspheres was achieved via covalent amide bond formation. The 12-PRV gE and 28-PRV gB microsphere complexes were resuspended in PBS–TBN (PBS, 0.1%BSA, 0.02% Tween-20, 0.05%Na_3_N_3_, pH 7.4) solution and stored at 4 °C in the dark. 

The specific operation steps are as follows: (1) the microsphere in the whole tube was vibrated with a vortex shaker for 1 min, and the microspheres precipitated at the bottom of the centrifugal tube were dispersed in the liquid to obtain uniform microsphere suspension; (2) an appropriate amount of microsphere suspension was taken into a 1.5 mL centrifuge tube, centrifuged at 8000× *g* for 2 min, placed on the magnetic frame, stood for 1 min, and the supernatant was aspirated with a pipette gun; (3) 100 µL ddH_2_O was added, vortexed for 1 min, 8000× *g*, 2 min centrifugation, placed on the magnetic frame, stood for 1 min, and the supernatant was aspirated with a pipette gun; (4) 80 µL 100 mM sodium dihydrogen phosphate solution (NaH2PO4) with pH = 6.2 was added, vortexed for 1 min, and the microspheres were resuspended; (5) 10 µL 50 mg/mL N-hydroxysulfosuccinide (n-sulfo-NHS) and 10 µL 50 mg/mL 1-ethyl-3 (3- (dimethylaminopropyl) carbodiimide) (EDC) were added for 1 min and; (6) incubated at room temperature for 20 min (gently vibrated with vortex shaker every 10 min), centrifuged at 8000× *g* for 2 min, placed on a magnetic frame, stood for 1 min, and the supernatant was absorbed with a pipette gun; (7) 250 µL 50 mm 2- (N-morpholino) ethanesulfonic acid (MES) with pH = 5.0 was added, vortexed for 1 min, placed on the magnetic frame, stood for 1 min, and the supernatant was aspirated with a pipette gun; repeated once. (8) 100 µL, and 50 mm pH = 5.0 MES were added into the activated microspheres, and the vortices were whirled for 1 min. Then, 100 µg of PRV gE and gB recombinant proteins were added into the activated microspheres as antigens. Finally, the volume was fixed to 500 µL, and the mixture was stirred by vortex; (9) it was placed on a shaker, incubated at room temperature for 2 h, centrifuged at 8000× *g* for 2–3 min, placed on a magnetic frame, stood for 1 min, and the supernatant was aspirated with a pipette gun; (10) 500 µL PBS–TBN was added, vortexed for 1 min, incubated on the shaker at room temperature for 30 min, underwent 8000× *g* centrifugation for 2–3 min, placed on the magnetic frame, stood for 1 min, and the supernatant was aspirated with a pipette gun; (11) then, 1 mL PBS–TBN was added, underwent 8000× *g* centrifugation for 2–3 min, the microspheres were precipitated, the supernatant discarded, and this was repeated again. The 12-PRV gE and 28-PRV gB microsphere complexes were resuspended in PBS–TBN (PBS, 0.1%BSA, 0.02% Tween-20, 0.05%Na_3_N_3_, pH 7.4) solution and stored at 4 °C in the dark. 

### 2.4. Selection of Optimal Serum Dilution

PRV gE/gB-positive and gE/gB-negative sera were diluted with PBS-1% BSA 10-, 20-, 40-, 80-, 160-, and 320-fold. We added 50 µL (250 µg/mL) of microspheres (50 beads/µL) and 50 µL of the sample to be tested (monoclonal antibody or serum) to each well in a 96-well plate and incubated the plate at room temperature for 1 h. Then, 100 µL of biotin-labeled rabbit anti-porcine IgG antibody (250 µg/mL) was added to each well, followed by the addition of 100 µL streptavidin-algae Red albumin (SA-PE, 1:1000) and incubated for 30 min at room temperature (500 g/min). Finally, the products were resuspended in 125 µL of sheath liquid per well and median fluorescent intensity (MFI) was read using a Luminex Flex 3D liquid phase detection system. The optimum serum dilution was determined based on the MFI values for each group.

### 2.5. Establishment and Evaluation of Dual FMIA

Using the Luminex xMAP system, gE-12 and gB-28 magnetic microbeads were diluted to 50 beads/µL using PBS–TBN, and 50 µL of gE-12 and 50 µL of gB-28 were added to each well of a 96-well microplate. After washing with PBS three times, 50 µL of the experimental sample was added to each well after incubation for 30 min at room temperature. Then, 100 µL of biotin-labeled rabbit anti-porcine IgG antibody (250 µg/mL) was added to each well, followed by the addition of 100 µL streptavidin-algae Red albumin (SA-PE, 1:1000) and incubated for 30 min at room temperature (500 g/min). Finally, the products were resuspended in 125 µL of sheath liquid per well and MFI was read using a Luminex Flex 3D liquid phase detection system. The above is the single FMIA step, and the double FMIA step is to add gE and gB experimental samples in the same hole and follow the same procedure. Ninety-six pig pathogen serum samples were tested using each single and dual FMIA, and the FMI values were assessed to evaluate the comparability of the two assays. The correlation coefficients were analyzed via linear regression analysis.

### 2.6. Comparison with ELISA

PRV gE and gB antibodies in pig sera were tested using the FMIA and the results were compared with those from an IDEXX ELISA kit. The MFI values were analyzed via receiver operating characteristic (ROC) curves using the MedCalc software to determine the optimal criterion value, specificity, and sensitivity. The chi-square test was used to evaluate the association [31,32].

### 2.7. Statistical Analysis

Experimental results were analyzed by one-way ANOVA or the Student’s *t*-test using SPSS software (version 21.0). The results are presented as the mean ±SD and a *p* < 0.05 was considered statistically significant. The specificity, coupling efficiency and serum dilution were evaluated using GraphPad Prism 6.0. In addition, Pearson’s correlation coefficient was calculated using the same software for the determination of relative coupling efficiencies and for the comparison of singleplex versus multiplex assays [33].

## 3. Results

### 3.1. Expression and Verification of Recombinant PRV gE and gB Proteins

To develop a dual detection method, we first amplified the gE aa 52–238 domain and gB aa 539–741 domain and constructed two recombinant plasmids, pMAL-c5x PRV gE and pMAL-c5x PRV gB. The two recombinant plasmids were identified via sequencing and the correct size was verified by PCR. The two expression plasmids were transfected into *Escherichia coli* BL21 (DE3). Extracts of pMAL-c5x PRV gE- and pMAL-c5x PRV gB-transformed cells were analyzed by SDS-PAGE and stained with Coomassie Brilliant Blue after induction with IPTG. The pMAL-c5x vector was used as a negative control. The results showed obvious bands of molecular mass, 63.3 and 66.3kDa, that corresponded to the PRV gE and gB recombinant proteins (including vector pMAL-c5x (added 6×His tag) at 43.3 kDa) (Figure 1A,B). PRV gE and gB recombinant proteins were abundant in the supernatant and consistent with the expected size of the recombinant tagged proteins, which demonstrated that the recombinant proteins were well expressed.

To verify that the recombinant proteins were the desired targets, supernatants were analyzed by immunoblotting with the primary antibodies (anti-MBP mAb and anti-His mAb) and secondary antibody (mouse IgG (H + L) goat anti-mouse antibody) after ultrasonic decomposition. The specific bands obtained were consistent with the size of the target proteins (Figure 2), indicating that the MBP-tagged and His-tagged recombinant proteins were successfully expressed.

### 3.2. Purification and Verification of the Antigenicity of the Recombinant Proteins

To develop an accurate FMIA method for detection, we purified the recombinant proteins. Supernatants of pMAL-c5x PRV gE and pMAL-c5x PRV gB from lysed cells were purified using Ni-Sepharose chromatography and the eluents were analyzed via SDS-PAGE and stained with Coomassie Brilliant Blue. The highest concentration of protein was eluted with 300 mM imidazole solution for gE (Figure 3A) and 200 mM imidazole solution for gB (Figure 3B). The two purified proteins were analyzed via immunoblotting with PRV-positive serum, which showed that the two fusion proteins reacted specifically with the corresponding serum (Figure 4A,C). No specific band appeared for the negative sera (Figure 4B,D). This demonstrated that the two purified recombinant proteins were suitable for developing an FMIA method based on good reactivity and antigenicity.

### 3.3. Evaluation of Specificity, Coupling Efficiency, Serum Dilution and Determination of the Optimal Concentration of Recombinant Protein Coupling

The purified recombinant proteins were analyzed for their coupling efficiency. gE-12 and gB-28 microsphere complexes obtained from the coupling of recombinant PRV gE and gB proteins with Number 12 and 28 magnetic microbeads were detected (Figure 5A). Compared with the negative serum, strong median fluorescent intensity (MFI) signals were detected in the serum positive for antigen-targeted gB and gE. This indicated that the coupling between PRV gE/gB and the magnetic microbeads was high enough for FMIA detection, and that the PRV gB and gE antigens could be used to detect antibodies in pig serum.

To select the optimal serum dilution, PRV gE/gB positive and negative sera were diluted 10-, 20-, 40-, 80-, 160-, and 320-fold with PBS. After fluorescence analysis, the MFI signals were >15,000 for the 10-fold dilution for PRV gE/gB positive serum, while the MFI signals for PRV gE/gB negative serum were <3000 (Figure 5B,C). The results showed that the optimal serum dilution ratio was 10 times.

The high specificity of single FMIAs is required for the development of a dual FMIA test, so we detected PRV gE/gB positive sera using single FMIAs, and the sera positive for other common pig pathogens including classical swine fever virus (CSFV), porcine circovirus (PCV), and porcine reproductive and respiratory syndrome virus (PRRSV), as well as PRV-negative serum, served as controls (Figure 6A). The results showed that there was no cross-reaction with these sera. For PRV gE IgG antibody detection, there were high MFI signals in the corresponding gE-positive sera, and similar results were obtained for PRV gB IgG antibody detection (Figure 6B). The results confirm that this single FMIA method had strong specificity.

The purified protein (gE/gB) used in each microsphere coupling reaction was 100 µg/mL. The purified recombinant gB protein was diluted 10 times from the initial concentration of 5 mg/mL, and the purified recombinant gE protein was diluted 10 times from the initial concentration of 10.8 mg/mL. The optimal protein concentration was determined by FMIA detection method. When the recombinant protein concentration was 100 µg/mL, the MFI of both gE and gB reached more than 20,000 (Figure 7).

### 3.4. Establishment of PRV gE and gB FMIA

Coupled gE-12 and gB-28 microsphere complexes were added to the same detection reaction to establish a dual detection method for gE and gB IgG. To evaluate the accuracy of this dual FMIA, we compared the MFI values for gE/gB single and duplex assays in our analyses of pig sera. Our results showed that the MFI values from the single assays were similar to those for the dual assay (Figure 8A,B). Thus, there was little cross-influence between the gE-12 and gB-28 microsphere complexes in the detection of antibodies in pig sera.

### 3.5. Comparison of ELISA and FMIA

To evaluate the superiority of our dual FMIA, we compared the results with the gE and gB ELISA test results. The serum collected by us has a clear background and was tested in our laboratory. The source of the PRV gB positive serum: the first immunization with PRV gE gene deletion live virus vaccine was performed in the pig farm, and the second immunization with PRV gE deletion inactivated vaccine was performed after 30 days, and the serum was collected after 14 days. Then, it was confirmed to be positive by indirect immunofluorescence assay. The source of the PRV gE serum: serum from different PRV-infected pig farms was collected and gE PCR was performed on the lymph nodes of corresponding pigs. The serum of the pigs with the positive results of gE-PCR was considered to be gE antibody positive. The source of the negative serum: the serum was collected for the immunofluorescence assay (IFA) test in pig farms that were not immunized with the PRV vaccine, and negative results were considered as negative samples. The serum samples used in the experiment: 254 gE positive and 30 gE negative sera; 379 gB positive serum and 30 gB negative serum. Receiver operating characteristic (ROC) curve analysis revealed that the criterion value for the optimal FMIA detection of gE antibodies was 5991.5. The sensitivity was 92.3% (95% confidence interval (CI) 63.9–98.7%), and the specificity was 99.26% (95% CI 97.4–99.9%) (Figure 9A). The area under the curve was 0.981 (*p* < 0.001) (Figure 9B). Similarly, the criterion value for the optimal FMIA detection of gB antibodies was 2862. For 409 gB clinical sera, the sensitivity was 96.3% (95% CI 81–99.4%) and the specificity was 95.74% (95% CI 89.5–98.8%) (Figure 9C). The area under the curve was 0.989 (*p* < 0.001) (Figure 9D). We used IDEXX ELISA method to analyze the ROC curve of the test results, and obtained the ROC curve (Figure 10A,C). The results showed that the cut-off value of gB ELISA was 0.399, the sensitivity was 95.83%, and the specificity was 91.37% (Figure 10B); the critical value of gE ELISA was 0.3191, the sensitivity was 91.7%, and the specificity was 88.89% (Figure 10D).The results demonstrate good correlation between the FMIA detection of gE/gB IgG antibodies and the IDEXX ELISA.

#### 3.5.1. Comparison of ELISA and FMIA Chi-Squared Tests

Ninety-two serum samples of gE and gB were selected for the FMIA test. The same samples were tested with EDEX ELISA kit. The number of positive sera and negative sera of FMIA and ELISA was calculated. Chi square test was performed on the two methods, and a statistical analysis was conducted. Compared with ELISA, and the positive and negative results of the FMIA in the clinical serum samples (Table 1 and Table 2), the correlation between the two methods was very significant, and both methods can reflect the same index.

#### 3.5.2. Repetitive Experiment

Intra-assay repeatability: two positive and one negative serum of PRV gE and gB were taken, with 10 replicates in each serum. The intra-assay precision of the method was evaluated by calculating the coefficient of variation. Inter-assay repeat: 11 positive and 1 negative serum of the PRV gE and gB were taken respectively, with three replicates in each serum. The inter-assay precision of the method was evaluated by calculating the coefficient of variation. The inter-assay precision of the method was evaluated by calculating the coefficient of variation. The average intra-assay coefficient of variation (CV) of gE and gB fluorescent microsphere immunoassay were 4.9% and 6.0% (Table 3 and Table 4), respectively, and the average inter-assay CV were 6.4% and 6.7% (Table 5 and Table 6). According to the requirements of Luminex’s xMAP^®^ Technical compendium, the intra-assay CV was no more than 10%, and the inter-assay CV was no more than 20%, indicating that the detection method established in this study has good repeatability.

## 4. Discussion

Pseudorabies has been prevalent on many pig farms in China since the 1970s, causing serious economic losses to the industry. In order to effectively control and eradicate PR, it is necessary to vaccinate with the gE gene deleted vaccine and carry out timely serological tests. On the one hand, we should ensure the level of gB antibody, and on the other hand, eliminate the pig with positive gE antibody. A gE-deleted vaccine, constructed via the homologous recombination or CRISPR/Cas9, provides safe and consistent protection in piglets and sows [34,35]. Disease surveillance is considered to be a complicated, time-consuming, and expensive process. To date, some commercial ELISA kits have been used to detect PRV gB or gE IgG antibodies in pigs [29,30].

In recent years, FMIAs have been used to diagnose human and animal pathogens. This method has also been used for differentiating and detecting yeast [36] and antibodies against pneumococcal polysaccharides and *Erysipelothrix rhusiopathiae* [37]. For the rapid detection of PRV, fluorescent immunochromatographic strips and indirect sandwich ELISAs were used [29,38], but there is no FMIA method for the detection of PRV IgG. In our study, the expression of recombinant antigens was the basis for the development of an immunoassay with high specificity and sensitivity. gE-deleted vaccines are used in many countries, and as one of the most conserved herpesvirus glycoproteins, gB can cause a host response. 

We selected gE and gB as the antigens to develop our FMIA for PRV detection. PRV gE (aa 52–238) and gB (aa 539–741) fragments [14,15], which contain major epitopes, were expressed in *E. coli* using the pMAL-c5X vector to obtain recombinant proteins. After purification and verification, the two recombinant proteins were coupled with Number 12 and 28 magnetic microbeads to form gE-12 and gB-28 microsphere complexes, respectively. These were used as substrates to verify the feasibility of separate FMIAs for detecting gE and gB antibodies in positive and negative sera. A dual FMIA for the detection of PRV gE and gB IgG antibodies was developed by integrating the single FMIAs. We evaluated the specificity of the dual FMIA using 1000 pig serum samples positive for PRRSV, PCV, and CFSV. The results demonstrated that the dual FMIA has strong specificity and the coupled antigens did not cross-react with other pig virus antibodies. The dual FMIA was compared with a commercial gE/gB ELISA kit (IDEXX Laboratories) to determine its accuracy. For gE detection IgG, the area under the ROC curve was 0.981, indicating that the dual FMIA has good accuracy for gE detection with high sensitivity (92.3%) and specificity (99.26%). The area under the curve was 0.989 for gB IgG detection, with high sensitivity (96.3%) and specificity (95.74%), confirming the accuracy of the dual FMIA for the detection of gB. The dual FMIA detection results showed good concordance and compatibility with gE/gB ELISA results. In addition, our dual FMIA method has the advantage that it can test several targets simultaneously, so it could be used for detecting several PRV targets in one sample.

FMIA is a novel immunoassay method that requires smaller sample volumes, less time, and is thus more economical than traditional ELISAs [37,39]. This method overcomes the ELISA disadvantage of having to analyze antigens separately. Compared with the commercial PRV IgG ELISA kit, our dual FMIA had higher sensitivity. FMIA can detect multiple antibodies in pig sera and provides an important method for the surveillance of pig diseases [40,41]. With the optimization of the reaction conditions, the multi-detection capability of FMIA provides a novel high-throughput and high-sensitivity detection platform for diagnosis of pathogens, monitoring of disease, and epidemiological investigations.

Taken together, we constructed a dual FMIA method by expressing the PRV gE and gB recombinant proteins and coupling them to fluorescent microspheres. Compared with a commercial ELISA kit, our dual FMIA method showed higher specificity and sensitivity. This provides a new method for monitoring PRV protective antibodies in vaccinated pigs and differentiating wild-type PRV infection from vaccinated responses simultaneously.

## Figures and Tables

**Figure 1 viruses-12-00912-f001:**
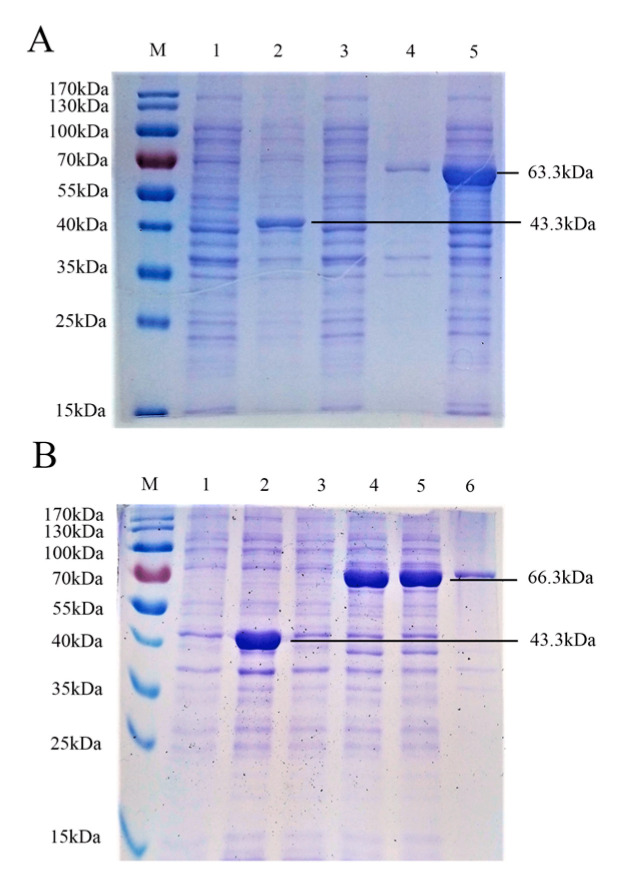
Identification of pseudorabies virus (PRV) gE and gB proteins from the pMAL-c5x plasmid via SDS-PAGE. (**A**) M, molecular weight marker; lane 1, untreated *E. coli* pMAL-c5x; lane 2, *E. coli* pMAL-c5x after induction by IPTG; lane 3, untreated *E. coli* pMAL-c5x PRV gE; lane 4, precipitate of *E. coli* pMAL-c5x PRV gE after induction by IPTG and ultrasonic disruption; lane 5, supernatant of *E. coli* pMAL-c5x PRV gE after induction by IPTG and ultrasonic disruption. (**B**) M, molecular weight marker; lane 1, untreated *E. coli* pMAL-c5x; lane 2, *E. coli* pMAL-c5x after induction by IPTG; lane 3, untreated *E. coli* pMAL-c5x PRV gB; lane 4, precipitate of *E. coli* pMAL-c5x PRV gB after induction by IPTG; lane 5, supernatant of *E. coli* pMAL-c5x PRV gB after induction by IPTG and ultrasonic disruption; lane 6, precipitate of *E. coli* pMAL-c5x PRV gB after induction by IPTG and ultrasonic disruption.

**Figure 2 viruses-12-00912-f002:**
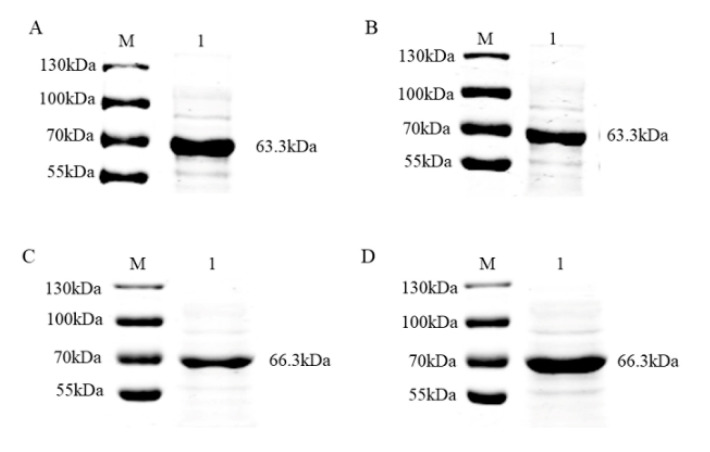
Identification of PRV gE and gB proteins from the pMAL-c5x plasmid via immunoblotting. (**A**) Detection of PRV gE by His tagging. M, molecular weight marker; lane 1, supernatant of *E. coli* pMAL-c5x PRV gE after induction by IPTG and ultrasonic disruption. (**B**) Detection of PRV gE by MBP (Maltose binding protein) tagging. M, molecular weight marker; lane 1, supernatant of *E. coli* pMAL-c5x PRV gE after induction by IPTG and ultrasonic disruption. (**C**) Detection of PRV gB by His tagging. M, molecular weight marker; lane 1, supernatant of *E. coli* pMAL-c5x PRV gB after induction by IPTG and ultrasonic disruption. (**D**) Detection of PRV gB by MBP tagging. M, molecular weight marker; lane 1, supernatant of *E. coli* pMAL-c5x PRV gB after induction by IPTG and ultrasonic disruption.

**Figure 3 viruses-12-00912-f003:**
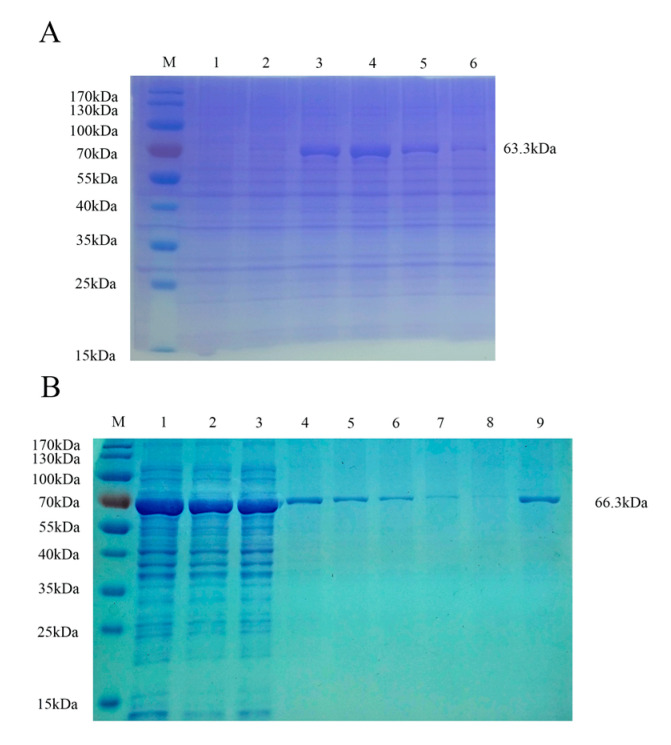
Identification of PRV gE and gB recombinant proteins after purification by SDS-PAGE. (**A**) Purification of PRV gE recombinant protein washed with different concentrations of imidazole solution and analyzed by SDS-PAGE. M, molecular weight marker; lane 1, recombinant PRV gE eluted with 80 mM imidazole solution; lane 2, recombinant PRV gE eluted with 100 mM imidazole solution; lane 3, recombinant PRV gE eluted with 200 mM imidazole solution; lane 4, recombinant PRV gE eluted with 300 mM imidazole solution; lane 5, recombinant PRV gE eluted with 400 mM imidazole solution; lane 6, recombinant PRV gE eluted with 500 mM imidazole solution. (**B**) Purification of PRV gB recombinant protein washed with different concentrations of imidazole solution and analyzed by SDS-PAGE. M, molecular weight marker; lane 1, supernatant of recombinant PRV gB after induction by IPTG and ultrasonic disruption; lane 2, recombinant PRV gB eluted with 100 mM imidazole solution; lanes 3 and 4, recombinant PRV gB eluted with 20 mM imidazole solution; lanes 5 and 6, recombinant PRV gB eluted with 40 mM imidazole solution; lanes 7 and 8, recombinant PRV gB eluted with 60 mM imidazole solution; lane 9, recombinant PRV gB eluted with 200 mM imidazole solution.

**Figure 4 viruses-12-00912-f004:**
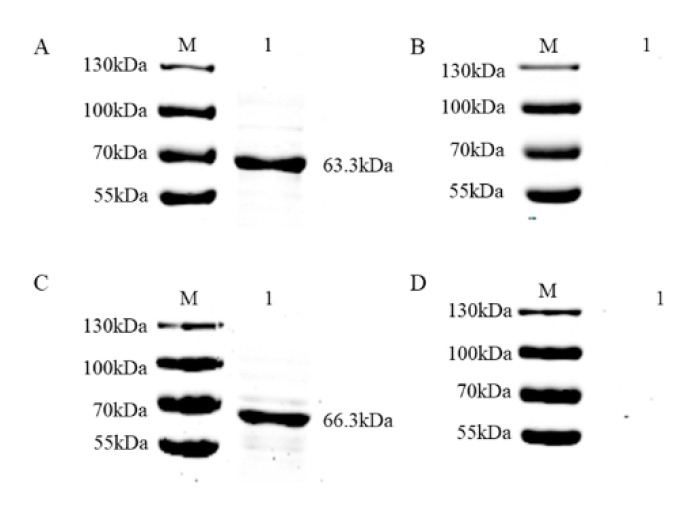
Antigenicity verification of PRV gE and gB recombinant proteins. PRV gE recombinant proteins were detected via immunoblotting using: (**A**) PRV gE-positive serum and (**B**) pig negative serum as the antibodies. PRV gB recombinant proteins were detected via immunoblotting using (**C**) PRV gB-positive serum and (**D**) pig negative serum as the antibodies.

**Figure 5 viruses-12-00912-f005:**
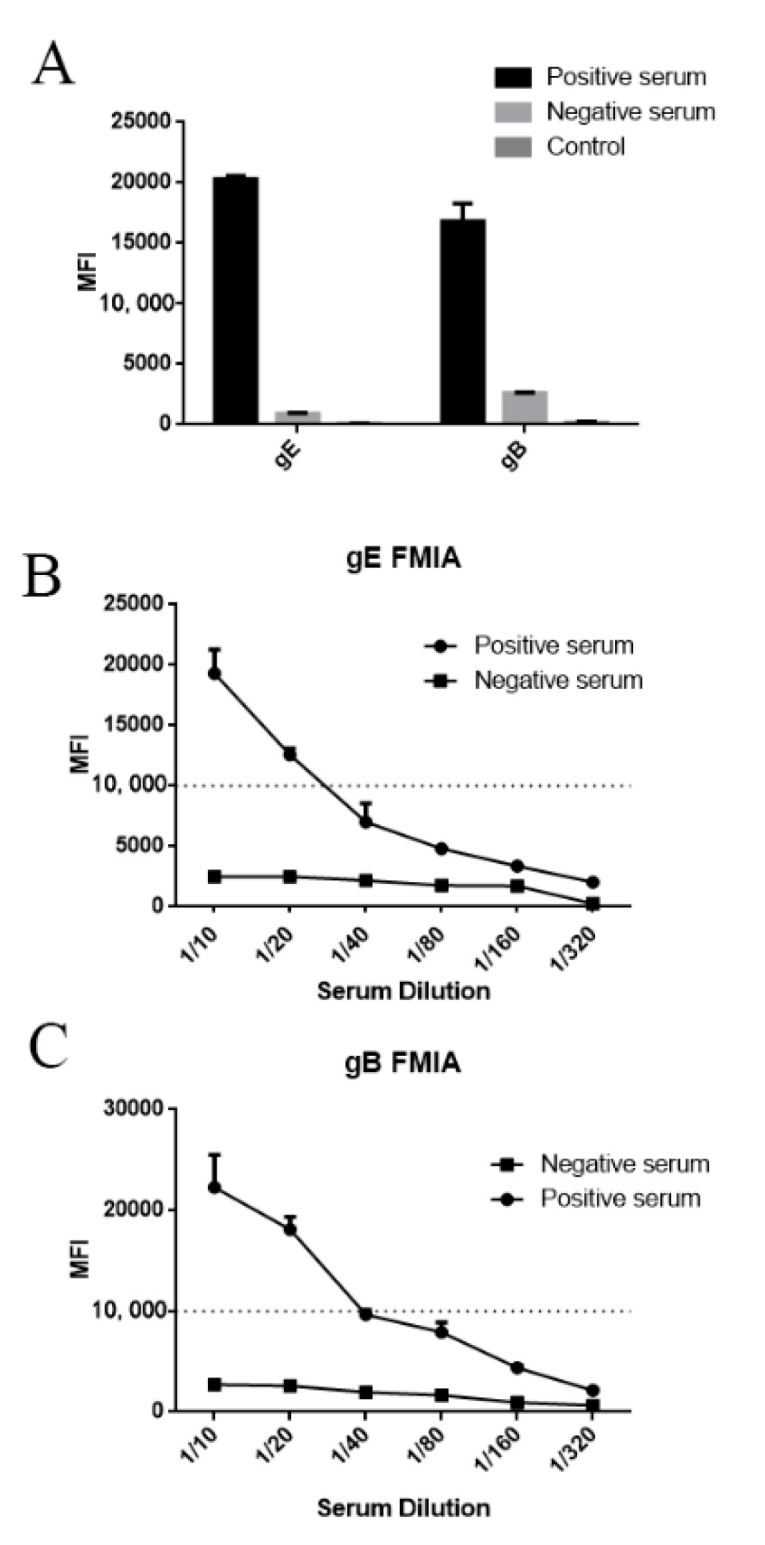
Median fluorescent intensity (MFI) value for PRV gE/gB positive and negative sera: (**A**) the MFI value of the fluorescent-microsphere immunoassays (FMIAs) was determined to test the PRV gE, and gB positive/negative sera were detected for feasibility analysis. (**B**,**C**) Samples were diluted 10-, 20-, 40-, 80-, 160-, and 320-fold using PBS, and the MFI value of FMIA was determined. At the same time, the optimal serum dilution was determined.

**Figure 6 viruses-12-00912-f006:**
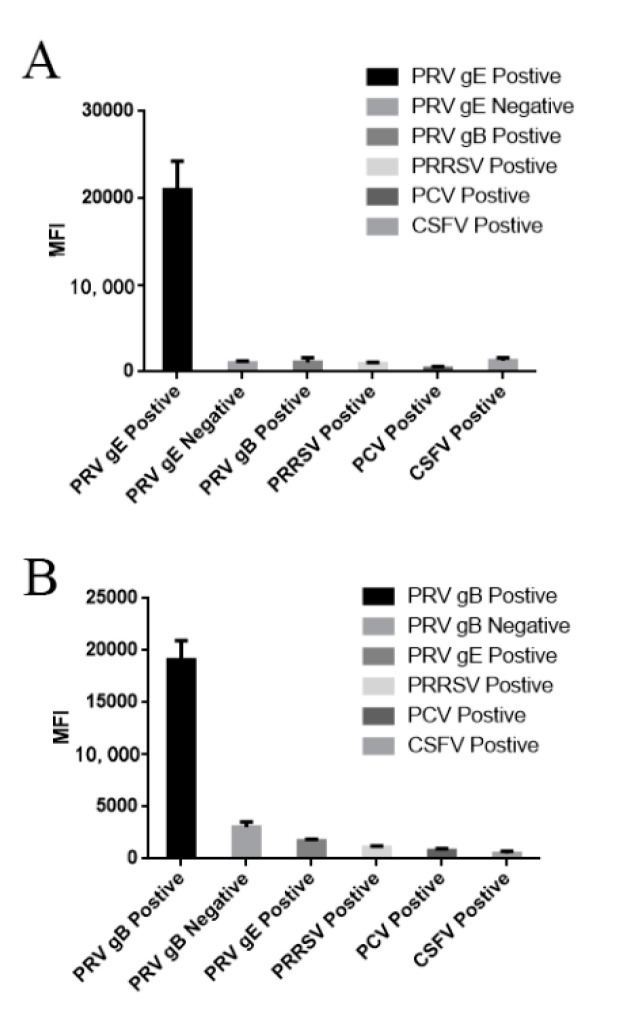
Antigenic cross-reactivity detection by dual FMIA: (**A**) the MFI value using the PRV gE IgG method to detect gE-positive serum was >20,000, while other common pig pathogens and negative serum had low MFI values; and (**B**) the MFI value using the PRV gB IgG method to detect gB-positive serum was >18,000, while other common pig pathogens and negative serum had low MFI values. PCV (porcine circovirus), CSFV (classical swine fever virus), PRRSV (Porcine Reproductive and Respiratory syndrome virus).

**Figure 7 viruses-12-00912-f007:**
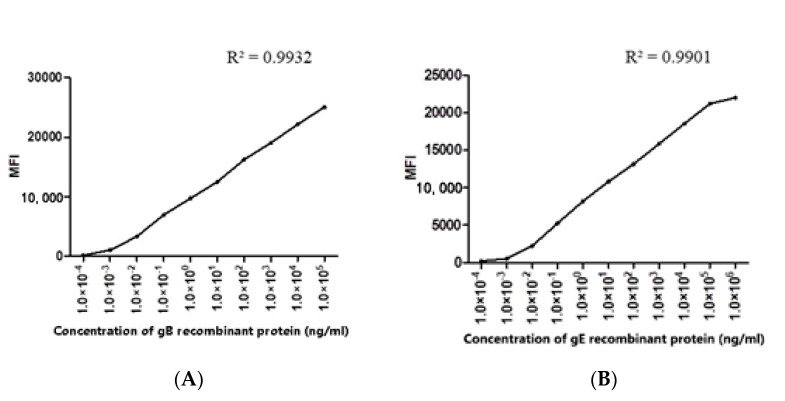
The optimal concentration of the recombinant protein was determined: (**A**) the determination of the optimal concentration of gB recombinant protein coupled with microspheres; (**B**) the determination of the optimal concentration of gE recombinant protein coupled with microspheres.

**Figure 8 viruses-12-00912-f008:**
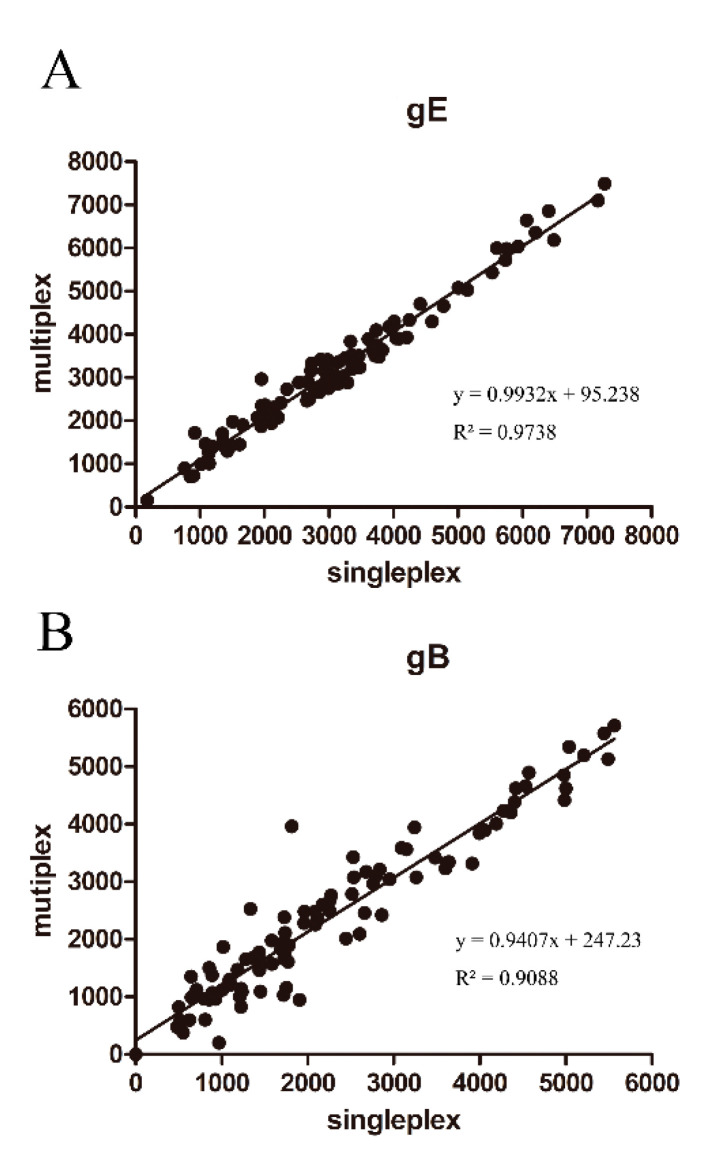
Evaluation of the dual FMIA detection methods: the comparison of MFI values detected from single and multiplex assays for (**A**) the PRV gE IgG detection and (**B**) the PRV gB IgG detection.

**Figure 9 viruses-12-00912-f009:**
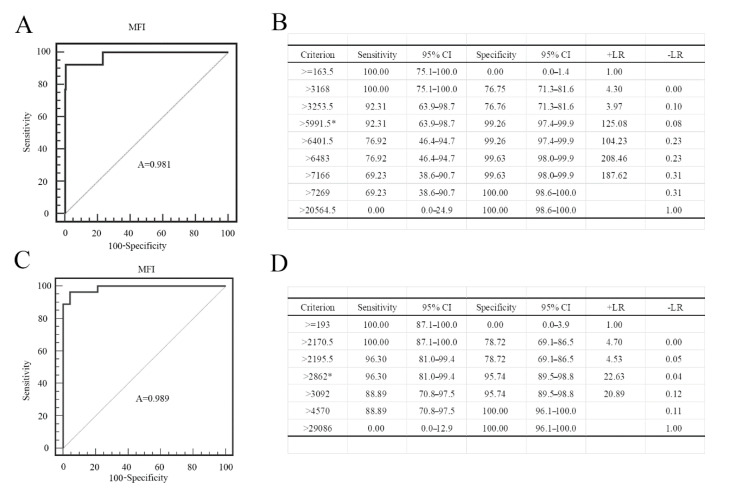
Statistical analysis of the PRV gE and gB FMIA results. Reactivity of the PRV gE sera according to (**A**) the receiver operating characteristic (ROC) curve analysis with (**B**) the criterion value (MFI > 5991.5 is positive and MFI < 5991.5 is negative) of the assay (5991.5) denoted by *. Reactivity of the PRV gB sera according to (**C**) the ROC curve analysis with (**D**) the criterion value of the assay (2862) denoted by *. We used the SigmaPlot 10.0 for the analysis of the sensitivity and specificity of the assay. +LR is positive likelihood ratio; −LR is negative likelihood ratio.

**Figure 10 viruses-12-00912-f010:**
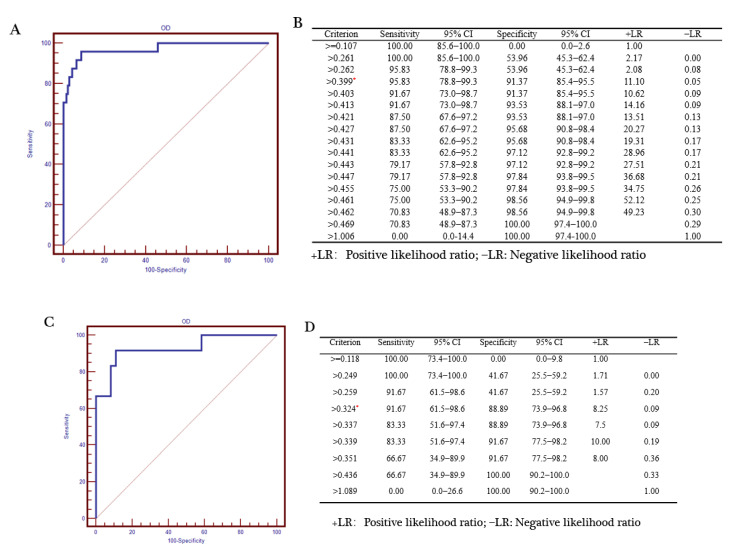
Statistical analysis of PRV gB and gE ELISA results. Reactivity of the PRV gB sera according to (**A**) the ROC curve analysis with (**B**) the criterion value (OD > 0.399 is positive and OD < 0.399 is negative) of the assay (0.399) denoted by *. Reactivity of the PRV gE sera according to (**C**) the ROC curve analysis with (**D**) the criterion value of the assay (0.324) denoted by *. The results showed that the cut-off value of gB ELISA was 0.399, the sensitivity was 95.83%, and the specificity was 91.37% (Figure 10B); the critical value of gE ELISA was 0.3191, the sensitivity was 91.7%, and the specificity was 88.89% (Figure 10D). We used SigmaPlot 10.0 for the analysis of the sensitivity and specificity of the assay. +LR is positive likelihood ratio; −LR is negative likelihood ratio.

**Table 1 viruses-12-00912-t001:** Results of the PRV gE serum samples detected by FMIA and ELISA.

Paired Chi Square Test		ELISA		
		+	−	Total
FMIA	+	33	1	34
	−	1	57	58
	total	34	58	92
Statistical results	Correlation test		Superiority test	
	Chi square value	*p*	Chi square value	*p*
	79.57	0.000	0.50	0.480
	Relevance significance		: there was no significant difference in advantage	

Note: paired chi square test (correlation chi square test, dominance chi square test).

**Table 2 viruses-12-00912-t002:** Results of the PRV gB serum samples detected by FMIA and ELISA.

Paired Chi Square Test		ELISA		
		+	−	Total
FMIA	+	32	2	34
	−	0	58	58
	total	32	60	92
Statistical results	Correlation test		Superiority test	
	Chi square value	*p*	Chi square value	*p*
	79.60	0.000	0.50	0.480
	Relevance significance		: there was no significant difference in advantage	

Note: paired chi square test (correlation chi square test, dominance chi square test).

**Table 3 viruses-12-00912-t003:** gE FMIA intra-assay repeat test.

Sample Number	$\bar{x}$ ± s(MFI)	CV%
1	491.55 ± 37.16	7.6
2	17,165.80 ± 456.90	2.7
3	25,527.70 ± 1139.50	4.4

**Table 4 viruses-12-00912-t004:** gB FMIA intra-assay repeat test.

Sample Number	$\bar{x}$ ± s(MFI)	CV%
1	420.20 ± 31.85	7.6
2	19,980.50 ± 516.40	2.6
3	14,108.9 ± 1089.7	7.8

**Table 5 viruses-12-00912-t005:** gE FMIA inter-batch repeat test.

Sample Number	$\bar{x}$ ± s(MFI)	CV%	Sample Number	$\bar{x}$ ± s(MFI)	CV%
1	6348.67 ± 199.80	3.1	7	11,071.10 ± 958.10	8.7
2	36,623.80 ± 1407.00	3.8	8	26,685.90 ± 2263.70	8.5
3	34,446.20 ± 1843.2	5.4	9	22,252.30 ± 501.40	2.3
4	10,220.17 ± 542.52	5.3	10	6835.00 ± 863.20	12.6
5	7480.00 ± 431.28	5.8	11	42,555.30 ± 3882.30	9.1
6	17,935.30 ± 1191.50	6.6	12	735.33 ± 41.02	5.6

**Table 6 viruses-12-00912-t006:** gB FMIA inter-batch repeat test.

Sample Number	$\bar{x}$ ± s(MFI)	CV%	Sample Number	$\bar{x}$ ± s(MFI)	CV%
1	17,935.30 ± 1191.50	6.6	7	26,685.90 ± 2263.70	8.5
2	10,273.33 ± 546.50	5.3	8	10,304.83 ± 1011.01	9.8
3	8949.17 ± 374.49	4.2	9	39,946.60 ± 2393.60	6.0
4	6685.67 ± 596.60	8.9	10	24,298.17 ± 2322.32	9.6
5	16,384.83 ± 915.12	5.6	11	8701.00 ± 474.37	5.5
6	31,847.90 ± 2571.4	8.1	12	450.00 ± 10.00	2.2

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
