# Peer review of "Development of a Dual Fluorescent Microsphere Immunological Assay for Detection of Pseudorabies Virus gE and gB IgG Antibodies"

_viruses, 2020, doi:10.3390/v12090912_

Round 1
Reviewer 1 Report
The article is concerned with the development of a serological test system based on the fluorescent microsphere technique which allows for simultaneous testing for field infection and vaccination titres of African Swine Fewer.
General comments:
I first believed this was going to be an easy review since abstract and introduction are very well done. But paper quality deteriorates with the specific elements in Material and Methods, Results etc.
I also wonder if the proposed application in disease control actually / practically makes sense:
Why should one test for vaccination titres in a routine disease control approach. Vaccine should be approved and vaccination recorded. DIVA-Strategy makes sense but testing for the deleted (in the vaccine) gE specific antibodies only would do the job. In routine-testing of non-suspicious animals the reported specificity of the gE component of the test (though relatively high) would produce a considerable amount of false-positives would be produced and the Positive Predictive Value would be way to poor if low ASF prevalence has to be assumed.
Anyway, for the sake of scientific advance I recommend the paper to be accepted for publication.
Most of the figures are too tiny – enlarge!
Major revisions:
Line 166: “conincidence rate”??? Do you mean “Association”?
Lines 165, 167, 169: I am not very happy with the description of the statistics. Under “Statistics” you should describe precisely which test was used for which hypothesis, e.g. when t-test / ANOVA etc.
You do not present the results of the ANOVA (probably used for the reactivity of different virus positive sera).
The Student`s t-test was used for what? Comparing results of dilution?
For comparison of FMIA and ELISA I guess a Kappa statistic and McNemar`s test would be more appropriate than the Chi-squared test – check with statistician!!!!
Line 252: Which statistical test? I assume, it`s significant also for dilutions 1:20 and higher.
Line 271 ff.: Please detail a little bit on the panel of sera used (96 – probably because of the number of wells) – positives and negatives? Positives of different reactivity? How did you choose??
Figure 5 A: Remove “B” next to legend. It´s slightly confusing that the “gE” / “gB” labels are positioned at tick marks under the negative sera. What was the control?
Figure 5 B: Remove “D”. Why a horizontal line at 10,000 (only)?
Figure 5 C: Why a horizontal line at 10,000 (only)?
Caption of Figure 5: Check logic / grammar – e.g. “(B,C) in wrong position!
Line281 ff.: I am struggling with the ROC analysis!
What are “positive clinical sera” – sera from clinically diseased animals?
The authors describe that they used sera positive in ELISA. So ELISA was considered the gold standard for the evaluation? How do you run ROC analysis with positive sera only?????
Lines 305-308: Sense in introducing this here????
Line 318 ff.: redundant / belongs to M & M!
Line 336 ff.: Whenever a higher sensitivity / specificity of the FMIA compared to ELISA is claimed -. How was this proven / where is this shown in the results???????????
Figure 8: Too small, explain last two columns of tables. Labeling of x-axis is wrong – should be “100 – Specificity”!
Minor revisions:
Line 142: In my printout a wrong symbol is displayed for 4°C.
Line 145: move “(beads / µl)” to position directly behind “microspheres”.
Lines 245 / 246: “serum positive for antigen-targeted gB and gE” ??? – serum positive for antibodies targeted on antigens gB and gE”?
Line 271: “Construction”? – maybe better: “setup”
Line 301: “of antigens”??? - of the disease (antibodies to the disease). Or do we talk about antigen-capture assays???? I don’t think so!
Line 303: “Virus surveillance” – better: “Disease surveillance”
Line 328: What’s “gE detection IgG”? Do you mean “for sera containing IgG specific to gE”?
Line 336: “requires fewer samples” – do you mean “smaller sample volumes”????
Author Response
Point 1: Line 166: “conincidence rate”??? Do you mean “Association”?
Response 1: It has been revised accordingly in Section 2.6.
Point 2: Lines 165, 167, 169: I am not very happy with the description of the statistics. Under “Statistics” you should describe precisely which test was used for which hypothesis, e.g. when t-test / ANOVA etc.
Response 2: I have added the related content in Section 3.5.1.
Point 3: You do not present the results of the ANOVA (probably used for the reactivity of different virus positive sera).
Response 3: I have added the related content in Section 3.5.1.
Point 4: The Student`s t-test was used for what? Comparing results of dilution?
Response 4: It's not a comparison of dilution results.I have added the related content in Section 3.5.2.
Point 5: For comparison of FMIA and ELISA I guess a Kappa statistic and McNemar`s test would be more appropriate than the Chi-squared test – check with statistician!!!!
Response 5: I have added the references in Section 2.6.
Point 6: Line 252: Which statistical test? I assume, it`s significant also for dilutions 1:20 and higher.
Response 6: We have made relevant modifications in this section.
Point 7: Line 271 ff.: Please detail a little bit on the panel of sera used (96 – probably because of the number of wells) – positives and negatives? Positives of different reactivity? How did you choose??
Response 7: Three different fluorescein were embedded in the microspheres of Flex Map 3D technology, and the ratio of each fluorescein was different, so there were 500 kinds of microspheres. Therefore, Flex Map 3D technology can detect 500 target molecules in the same sample at a time. Different groups such as carboxyl group, histamine group and tag sequence are linked on the microspheres. The groups can be combined with probe molecules (antigen, antibody and oligonucleotide probe) containing amino group, carboxyl group and tag nucleic acid group. After binding, the microspheres can be used as capture carrier, which can specifically bind with the target molecules in the sample to be detected, and then combine with the fluorescent reporter molecule of phycoerythrin to complete the reverse reaction The microsphere should be read on the machine. When reading, the instrument emits two laser beams to irradiate the microsphere. One red laser identifies the fluorescein of the microsphere itself and recognizes the code of the microsphere; the other laser identifies the fluorescence of phycoerythrin, and carries out quantitative analysis of the sample through the fluorescence intensity. It is only effective when both laser beams detect the signal, thus avoiding the interference of other substances. In addition, when one sample was detected, the fluorescence values of 100 microspheres were randomly detected by FMIA, and the median was taken as the median fluorescence intensity (MFI). Therefore, we can detect gE and gB simultaneously, which can be specifically recognized
Point 8: Figure 5 A: Remove “B” next to legend. It´s slightly confusing that the “gE” / “gB” labels are positioned at tick marks under the negative sera. What was the control?
Response 8: We have deleted B in Figure 5A, PRV gE / gB stands for PRV gE and PRV gB.
Point 9: Figure 5 B: Remove “D”. Why a horizontal line at 10,000 (only)?
Response 9: We have deleted D in Figure 5B. We put the horizontal line at 10000 because according to the technical requirements, the MFI (mean fluorescence intensities) of positive serum is greater than 10000, and it is more than 5 times of MFI of negative control, and the MFI of blank control is less than 100, which indicates that the coupling method and detection method are feasible.
Point 10: Figure 5 C: Why a horizontal line at 10,000 (only)?
Response 10: We put the horizontal line at 10000 because according to the technical requirements, the MFI (mean fluorescence intensities) of positive serum is greater than 10000, and it is more than 5 times of MFI of negative control, and the MFI of blank control is less than 100, which indicates that the coupling method and detection method are feasible.
Point 11: Caption of Figure 5: Check logic / grammar – e.g. “(B,C) in wrong position!
Response 11: We have modified it accordingly in Figure 5.
Point 12: Line281 ff.: I am struggling with the ROC analysis! What are “positive clinical sera” – sera from clinically diseased animals?
Response 12: Positive clinical serum refers to the serum from clinically PRV infected animals.
Point 13: The authors describe that they used sera positive in ELISA. So ELISA was considered the gold standard for the evaluation? How do you run ROC analysis with positive sera only?????
Response 13: The positive samples were confirmed to be positive by our test, so ELISA is not the gold standard. The positive and negative samples were detected by FMIA method, and ROC analysis was performed by medcalc software.
Point 14: Lines 305-308: Sense in introducing this here???? Line 318 ff.: redundant / belongs to M & M!
Response 14: We have removed these redundant statements in this section.
Point 15: Line 336 ff.: Whenever a higher sensitivity / specificity of the FMIA compared to ELISA is claimed -. How was this proven / where is this shown in the results???????????
Response 15: It has been revised accordingly in Section 3.5.
Point 16: Figure 8: Too small, explain last two columns of tables. Labeling of x-axis is wrong – should be “100 – Specificity”!
Response 16: We have enlarged Figure 8 and the X-axis has been modified. In the last two columns of the table, + LR represents positive likelihood ratio, and the value range of + LR is (0, ¥ ). The larger the value, the stronger the ability of the detection method to confirm the disease. -LR represents negative likelihood ratio, and the value range of - LR is (0, ¥). The smaller the value, the better the ability of the detection method to exclude diseases.
Point 17: Line 142: In my printout a wrong symbol is displayed for 4°C. Line 145: move “(beads / µl)” to position directly behind “microspheres”.
Response 17: It has been revised accordingly in Section 2.3 and 2.4.
Point 18: Lines 245 / 246: “serum positive for antigen-targeted gB and gE” ??? – serum positive for antibodies targeted on antigens gB and gE”?
Response 18: In order to verify whether the recombinant PRV gE and gB protein coupled with magnetic microspheres Number 12 and 28 can specifically recognize PRV, the positive serum of PRV gE and PRV gB was selected for detection.
Point 19: Line 271: “Construction”? – maybe better: “setup” .Line 301: “of antigens”??? - of the disease (antibodies to the disease). Or do we talk about antigen-capture assays???? I don’t think so!
Response 19: It has been revised accordingly in Section 3.4 and 4.
Point 20: Line 303: “Virus surveillance” – better: “Disease surveillance”. Line 328: What’s “gE detection IgG”? Do you mean “for sera containing IgG specific to gE”?
Response 20: This was modified accordingly in paragraph 1 of the discussion. Yes, the meaning of the article is for the sera containing IgG against gE.
Point 21: Line 336: “requires fewer samples” – do you mean “smaller sample volumes”????
Response 21: Yes, the meaning of the article is that the sample volumes is small.

Reviewer 2 Report
The central argument of this manuscript is the development of a dual fluorescent microsphere immunological assay for detection of pseudorabies virus gE and gB IgG antibodies. PRV causes huge economic losses in the pig industry especially due to the variant PRV strains that emerged in swine in China in 2011. Ji and his colleagues provided a new method for monitoring PRV protective antibodies (gE &gE) with high specificity and sensitivity in vaccinated pigs and differentiating wild-type PRV infection from vaccinated responses simultaneously.
Major comments and concerns:
Comment #1) The description of the coupling of the recombinant proteins to the Luminex microspheres is confusing. was each protein individually coupled to a bead set, after which the sets were mixed to create a dual FMI assay? If so, please clarify this in the manuscript.
Comment #2) Figure 1: what is the difference between lane #4 and lane #6 in panel 2 of this figure?? They look different for the protein expression although they are the same thing, (precipitate of E. coli pMAL-c5x PRV gB after induction by IPTG and ultrasonic disruption), please clarify.
Comment #3) Line 153-161: For establishment and evaluation of the dual FMIA, the method is not clear, and many steps are missed.
Comment #4) Line 178: I think there is a mistake, the authors mentioned that the two expression plasmids were transfected into Escherichia coli BL21 (DE3) but this is not compatible with what mentioned in the method part (Line 120), please clarify.
Comment #5) What was the quantity of the purified proteins (gE/gB) used in each microsphere coupling reaction?? Have you tested different quantities of purified recombinant proteins bound to beads to select the optimal protein concentration??
Comment #6) How can the authors judge the test sensitivity by 1/10 dilution is the optimal dilution and this is the lowest dilution compared with the other dilutions (1/20, 1/40, 1/80……..), why did not the authors start with dilutions 1/2, 1/4, 1/8,…..) to give the authors some space to determine the analytical sensitivity of the single and dual FMIA ???
Comment #7) Figure 7: Authors should provide an informative caption for this figure.
Comment #8) The discussion in the manuscript is not well written, I have noticed that it looks like the results. In the paragraph from line 318 to line 335, there is no citations. Also, in the last paragraph from line 336 to line 347, there is only one citation for the conclusion in the manuscript. There are many recent publications related to this work that the authors can cite in their discussion.
Minor comments:
Line 48-49: The family and subfamily names must be italic.
Line 50: Kbp instead of Kb because PRV is a double-stranded DNA virus.
Line 118-119: please provide the restriction enzymes (sites) used for sub-cloning of gB and gE coding genes in the vector.
Line 120: Have the authors checked the correct obtained constructs after subcloning??
Line 269: (gE-positive) should be gB-positive.
Line 279: (multiplex) should be replaced by dual or duplex to be compatible with what mentioned in the text.
Author Response
Point 1: The description of the coupling of the recombinant proteins to the Luminex microspheres is confusing. was each protein individually coupled to a bead set, after which the sets were mixed to create a dual FMI assay? If so, please clarify this in the manuscript.

Response 1: On behalf of all co-authors, I would like to take this opportunity to thank you for constructive suggestions and comments on our manuscript. It has been revised accordingly in Section 2.3 and 2.5.
Point 2: Figure 1: what is the difference between lane #4 and lane #6 in panel 2 of this figure?? They look different for the protein expression although they are the same thing, (precipitate of E. coli pMAL-c5x PRV gB after induction by IPTG and ultrasonic disruption), please clarify.
Response 2: lane 4, precipitate of E. coli pMAL-c5x PRV gB after induction by IPTG,lane 6, precipitate of E. coli pMAL-c5x PRV gB after induction by IPTG and ultrasonic disruption. It has been revised accordingly in Section 3.1.
Point 3: Line 153-161: For establishment and evaluation of the dual FMIA, the method is not clear, and many steps are missed.
Response 3: Response 15: It has been revised accordingly in Section 2.5.
Point 4: Line 178: I think there is a mistake, the authors mentioned that the two expression plasmids were transfected into Escherichia coli BL21 (DE3) but this is not compatible with what mentioned in the method part (Line 120), please clarify.
Response 4: Line 120 mentions E. coli DH5α as a clone of the recombinant plasmid and coli BL21 (DE3) as an induced expression of the recombinant plasmid.
Point 5: What was the quantity of the purified proteins (gE/gB) used in each microsphere coupling reaction?? Have you tested different quantities of purified recombinant proteins bound to beads to select the optimal protein concentration??
Response 5: The purified protein (gE / gB) used in each microsphere coupling reaction was 100 μg/mL. The purified recombinant gB protein was diluted 10 times from the initial concentration of 5 mg/ml, and the purified recombinant gE protein was diluted 10 times from the initial concentration of 10.8 mg/ml. the optimal protein concentration was determined by FMIA detection method. When the recombinant protein concentration was 100 μg/ml, the MFI of both gE and gB reached more than 20000.
Point 6: How can the authors judge the test sensitivity by 1/10 dilution is the optimal dilution and this is the lowest dilution compared with the other dilutions (1/20, 1/40, 1/80……..), why did not the authors start with dilutions 1/2, 1/4, 1/8,…..) to give the authors some space to determine the analytical sensitivity of the single and dual FMIA ???
Response 6: According to the results, when 1 / 10 serum is diluted, the MFI value detected by FMIA is the highest, which can be determined as the optimal dilution degree. We do not dilute from 1 / 2, 1 / 4, 1 / 8 because the serum concentration is too high to affect the detection accuracy.
Point 7: Comment #7) Figure 7: Authors should provide an informative caption for this figure.
Response 7: We have modified Figure 7 in the article as required. The title is: evaluation of dual FMIA detection methods.
Point 8: The discussion in the manuscript is not well written, I have noticed that it looks like the results. In the paragraph from line 318 to line 335, there is no citations. Also, in the last paragraph from line 336 to line 347, there is only one citation for the conclusion in the manuscript. There are many recent publications related to this work that the authors can cite in their discussion.
Response 8: I added references and made changes in Section 4.
Point 9: Line 48-49: The family and subfamily names must be italic.
Response 9: We have revised it in the paragraph 2 in the Introduction.
Point 10: Line 50: Kbp instead of Kb because PRV is a double-stranded DNA virus.
Response 10: We have revised it in the paragraph 2 in the Introduction.
Point 11: Line 118-119: please provide the restriction enzymes (sites) used for sub-cloning of gB and gE coding genes in the vector.
Response 11: Both recombinant plasmids were identified by restriction endonucleases EcoR I and hind Ш. I have added the related content in Section 2.2.
Point 12: Line 120: Have the authors checked the correct obtained constructs after subcloning??
Response 12: After obtaining the recombinant plasmids, two recombinant plasmids were identified by restriction endonucleases EcoR I and hind Ш. Gel electrophoresis results showed that the pMAL-c5x PRV gE enzyme cut obtained two bands of 5677 bp and 547 bp, which were consistent with the expected results (Figure. 1); pMAL-c5x PRV gB enzyme digestion obtained two bands of 5677 bp and 609 bp , which were consistent with the expected results (Figure. 2).
Figure. 1 Identification of recombinant plasmid pmal-c5x PRV gE
Note: M:DL5000 Marker; 1,2: enzyme digestion products
Figure. 2 Identification of recombinant plasmid pmal-c5x PRV gB
Note: M:DL5000 Marker;1:enzyme digestion products
Point 13: Line 269: (gE-positive) should be gB-positive.
Response 13: I've made the relevant changes on line 305.
Point 14: Line 279: (multiplex) should be replaced by dual or duplex to be compatible with what mentioned in the text.
Response 14: It has been revised accordingly in Section 3.4.
Please also check the attachment for figures

Round 2
Reviewer 1 Report
I appreciate the improvements of the paper, but there are still unacceptable flaws:
A "minor" one:
Line 341: 3.5.1 Comparison of ELISA and FMIA ANOVA
You carry out chi-squared tests, that's definitely not ANOVA
Major, and without appropriate explanation I will have to recommend rejection - The following aspect was not clarified:
Point 13: The authors describe that they used sera positive in ELISA. So ELISA was considered the gold standard for the evaluation? How do you run ROC analysis with positive sera only?????
Response 13: The positive samples were confirmed to be positive by our test, so ELISA is not the gold standard. The positive and negative samples were detected by FMIA method, and ROC analysis was performed by medcalc software.
The answer does not address the problem. Now you present a second ROC analysis for ELISA. The question remains:
How do you run ROC analysis with positive samples only????
Your statement:
We used the dual FIMA to test 284 gE- and 409 gB-positive clinical sera according to IDEXX ELISA.
So you used only positive sera (determined by ELISA) to evaluate your test - this cannot work, you need negative sera!
Against which test did you evaluate the ELISAs in the now added ROC analysis for ELISA???? Also sera positive in ELISA??? This does not work!
Maybe you used a panel of positive and negative sera as determined by any test, but than you have to describe this!!!
Author Response
Point 1: Line 341: 3.5.1 Comparison of ELISA and FMIA ANOVA.You carry out chi-squared tests, that's definitely not ANOVA.
Response 1: It has been revised accordingly in Section 3.5.1.
Point 2: The authors describe that they used sera positive in ELISA. So ELISA was considered the gold standard for the evaluation? How do you run ROC analysis with positive sera only?????
Response 2: On behalf of all co-authors, I would like to take this opportunity to thank you for constructive suggestions and comments on our manuscript. ELISA is not the gold standard. The serum collected by us has a clear background and is tested in our laboratory.
(1) Source of PRV gB positive serum: The first immunization with PRV gE gene deletion live virus vaccine was performed in the pig farm, and the second immunization with PRV gE deletion inactivated vaccine was performed after 30 days, and the serum was collected after 14 days. Then it was confirmed to be positive by indirect immunofluorescence assay.
(2) Source of PRV gE serum: Serum from different PRV-infected pig farms was collected and gE PCR was performed on lymph nodes of corresponding pig. The serum of the pigs with the positive results of gE-PCR was considered to be gE antibody positive.
(3) Source of negative serum: Serum was collected for IFA test in pig farms that were not immunized with PRV vaccine, and negative results were considered as negative samples.
Serum samples used in the experiment: 254 gE positive and 30 gE negative sera; 379 gB positive serum and 30 gB negative serum. It has been revised accordingly in Section 3.5.
Point 3: How do you run ROC analysis with positive samples only????
Response 3: We are sorry that we did not clearly describe in the article and there is information missing. When we drew the ROC curve, we used both positive and negative serum to confirm the specificity and sensitivity of the FMIA detection method. It has been revised accordingly in Section 3.5.
Point 4: Against which test did you evaluate the ELISAs in the now added ROC analysis for ELISA???? Also sera positive in ELISA??? This does not work!
Response 4: We used ELISA kit and FMIA to detect the same serum samples at the same time. ROC analysis was performed to confirm the specificity and sensitivity of ELISA and FMIA in detecting PRV gE and gB serum. Then the specificity and sensitivity of the two methods were compared according to the ROC curve analysis. Serum samples used in the experiment: 254 gE positive and 30 gE negative sera; 379 gB positive serum and 30 gB negative serum.

Reviewer 2 Report
I found that the authors revised their manuscript. All concerns and questions have been considered and addressed so the manuscript has been significantly improved.
I would like to suggest to include the diagram showing the result of testing different quantities of purified recombinant proteins as a supplementary figure for the manuscript.
Finally, I recommend the paper to be accepted for publication in Viruses.
Author Response
Point 1: I would like to suggest to include the diagram showing the result of testing different quantities of purified recombinant proteins as a supplementary figure for the manuscript.
Response 1: It has been revised accordingly in Section 3.3.

Round 3
Reviewer 1 Report
Seems to be ok now!